# Genome-wide identification and expression analysis of the phosphatase 2A family in rubber tree (*Hevea brasiliensis*)

**Jinquan Chao** [1], **Zhejun Huang**[2], **Shuguang Yang**[1], **Xiaomin Deng**[1], **Weimin Tian**[1]*

**1** Ministry of Agriculture and Rural Affairs Key Laboratory of Biology and Genetic Resources of Rubber Tree/ State Key Laboratory Breeding Base of Cultivation and Physiology for Tropical Crops, Rubber Research Institute, Chinese Academy of Tropical Agricultural Sciences, Haikou, Hainan, P. R. China, **2** College of Foresty, Hainan University, Haikou, Hainan, P. R. China

* wmtian@163.com

**Data Availability Statement:** All relevant data are within the paper and its Supporting Information files.

## Abstract

The protein phosphatase 2As (PP2As) play a key role in manipulating protein phosphorylation. Although a number of proteins in the latex of laticifers are phosphorylated during latex regeneration in rubber tree, information about the PP2A family is limited. In the present study, 36 members of the HbPP2A family were genome-wide identified. They were clustered into five subgroups: the subgroup HbPP2AA (4), HbPP2AB' (14), HbPP2AB'' (6), HbPP2AB55 (4), and HbPP2AC (8). The members within the same subgroup shared highly conserved gene structures and protein motifs. Most of *HbPP2As* possessed ethylene- and wounding-responsive *cis*-acting elements. The transcripts of 29 genes could be detected in latex by using published high-throughput sequencing data. Of the 29 genes, seventeen genes were significantly down-regulated while *HbPP2AA1-1* and *HbPP2AB55α/Bα-1* were up-regulated by tapping. Of the 17 genes, 14 genes were further significantly down-regulated by ethrel application. The down-regulated expression of a large number of *HbPP2As* may attribute to the enhanced phosphorylation of the proteins in latex from the tapped trees and the trees treated with ethrel application.

## Introduction

Protein phosphatase 2A (PP2A), a class of protein serine/threonine phosphatases, play a key role in manipulating protein phosphorylation by removing phosphate groups from the modified proteins [1–2]. The PP2A holoenzyme consists of three subunits: the structural subunit (A), the regulatory subunit (B), and the catalytic subunit (C) [3–4]. The subunit A comprises a series of conserved alpha-helical repeats, providing a scaffold for the binding of B and C; the subunit B determines substrate specificity; the subunit C dimerizes with A to form an active conformation and serves as a platform for interactions with B [5]. Based on their structural characteristics, the B subunits are further classified into B, B′, and B″ [6]. The large diversification of subunits confers on PP2A important roles in various biological processes [7]. Over-expression of *StPP2Ac2b* in *Solanum tuberosum* shows higher tuber induction rates compared

**Funding:** This research was partially supported by the National Key R&D Program of China (2018YFD1000502) and Modern Agro-industry Technology Research System (CARS-33-YZ1) awarded to WM.T. This research was also partially supported by the National Natural Science Foundation of China (31700601) awarded to J.C.

**Competing interests:** The authors have declared that no competing interests exist.

to wild-type stolons [8]. In maize, over-expression *ZmPP2AA1* improves low phosphate tolerance by remodeling the root system [9]. In Arabidopsis, the *pp2a-b′γ* and *pp2a-b′θ* mutants display lowered susceptibility towards the bacterial pathogen *Pseudomonas syringae* [10].

The rubber tree is the world's main source of natural rubber (*cis*-1, 4-polyisoprene) [11]. As an important industrial raw material, natural rubber is synthesized by the laticifers located in the inner bark of the rubber tree [12]. The milky laticifer cytoplasm, named latex, contains 20%-40% natural rubber. In natural rubber production, the milky latex is obtained by successive tapping, a process of artificially severing the laticifer rings every 2–3 days [13]. Latex exploitation by tapping enhances latex regeneration and the efficiency of natural rubber biosynthesis by activating jasmonate signalling in laticifers [14]. To increase the tapping efficiency, ethrel is applied to increase the rubber yield per tapping to economize on tapping workers. Prolongation of the duration of latex flow mainly increases the rubber yield per tapping by increasing the level of a 44 kDa protein in the cytosol of laticifer cells [15]. Ethrel itself, however, inhibits rubber biosynthesis [14].

Available data show that ethrel application induces the phosphorylation of a number of proteins in latex [16], which may be the mechanism behind changes in the efficiency of rubber biosynthesis. Although several protein kinase gene families such as mitogen-activated protein kinase (MPK) [17] and calcium-dependent protein kinase (CPK) [18–19] have been identified in rubber tree, information on the protein phosphatase gene families is limited. In the present study, 36 members of the *HbPP2A* gene family were genome-wide identified based on the latest version of the *H. brasiliensis* genome [20], and their structures and phylogenetics, and the ethrel-induced expression patterns were analyzed.

## Materials and methods

### Plant materials and treatments

Ten-year-old virgin trees and regularly tapped trees of the *H. brasiliensis* clone CATAS7-33-97 were grown on the Experimental Farm of the Chinese Academy of Tropical Agricultural Sciences (CATAS) on Hainan Island. The ten-year-old trees were regularly tapped using a half spiral pattern every 3 days (S/2, d/3). The latex samples from ten tapped trees or virgin trees were combined as one sample. For ethrel treatment, the regularly tapped rubber trees had 1.5% ethrel applied on the tapping panel, and we collected the latex samples after 24 h. Three biological replications were performed.

### RNA isolation and cDNA synthesis

Total RNA was extracted from samples using RNAplant Plus reagent (TIANGEN Biotech Co., Ltd., Beijing, China) and evaluated using a NanoDrop 2000 (Thermo Scientific Inc., USA). Approximately 1 μg of RNA was used for reverse transcription, and cDNA was synthesized using RevertAid™ First Strand cDNA Synthesis Kit (Thermo Scientific Inc., USA).

### Computational analysis

The whole-genome sequence of *H. brasiliensis* was downloaded from NCBI (https://www.ncbi.nlm.nih.gov/genome/?term=rubber%20tree%20genome). The published transcriptome data of seven rubber tree tissues were downloaded from the DRASearch database (http://trace.ddbj.nig.ac.jp/DRASearch/): root (SRR3136158), bark (SRR3136156), leaf (SRR3136159), male flower (SRR3136166), female flower (SRR3136165), latex (SRR3136162), seed (SRR3136168). Arabidopsis PP2A protein sequences were acquired from the TAIR database (http://www.arabidopsis.org/). The BLAST algorithm was also used to identify the predicted *H. brasiliensis*

PP2A proteins using AtPP2A as the query. All putative HbPP2A proteins were further confirmed by the CDD database (http://www.ncbi.nlm.nih.gov/cdd/). The AtPP2A and HbPP2A sequences were aligned using the online software Multiple Sequence Alignment (https://www.ebi.ac.uk/Tools/msa/), and then the phylogenetic tree was constructed using iTOL online software (http://itol.embl.de/). The molecular weight and isoelectric points of the HbPP2A proteins were predicted from the ExPASy database (http://expasy.org/). The gene structures and protein motif analyses were performed using GSDS2.0 software (http://gsds.cbi.pku.edu.cn/) and MEME software (http://meme-suite.org/tools/meme), respectively [21]. For MEME analysis, the number of motifs was set as 20. The identified motifs were annotated by the InterProScan database (http://www.ebi.ac.uk/Tools/pfa/iprscan/). The 1,500 bp genomic sequences upstream of the ATG of the *HbPP2A* genes were downloaded from the published *H. brasiliensis* genomes, and analyzed by PlantCARE (http://bioinformatics.psb.ugent.be/webtools/plantcare/html/) for *cis*-acting regulatory elements identification.

### qRT-PCR analysis

The primers for the *HbPP2A* genes were designed using the Primer Premier 5 software (S1 Table). The experiment was performed with the CFX384 real-time PCR system (Bio-Rad, USA) using the SYBR Prime Script RT-PCR Kit (TaKaRa, Dalian). *HbUBC2b* was used as the standard control for gene normalization [22]. Three biological replicates were measured.

### Statistical analysis

Changes in target gene relative expression levels across hormone treatments and tapping experiments were assessed by the $2^{-\Delta\Delta Ct}$ method. Expression level differences among different treatments were tested for significance by Duncan's test using SPSS Statistics 17.05. The capital letter or lowercase represented $p < 0.01$ or $p < 0.05$, respectively.

## Results

### Identification of the HbPP2A genes in rubber tree

Three AtPP2AA, nine AtPP2AB', six AtPP2AB", two AtPP2AB55, and five AtPP2AC genes were used as queries to perform genome-wide identification of the HbPP2A family in rubber tree based on tBLASTn searches (E<1−150). The HEAT domain (PF13646), the B56 domain (PF01603), the EF-hand domain (PF13499), and the WD40 domain (PF00400) in the putative members of the HbPP2A family were respectively further validated by using the NCBI Conserved Domain Search Service. In this way, thirty-six HbPP2A genes were identified and designated based on their presumptive Arabidopsis orthologue names. They included four *HbPP2AAs*, fourteen *HbPP2AB's*, six *HbPP2AB"s*, four *HbPP2AB55s*, and eight *HbPP2ACs*. The length of the HbPP2A proteins ranged from 280 aa (HbPP2AC6) to 588 aa (HbPP2AA1-1 and HbPP2AA2), the molecular weights ranged from 31.56 kDa (HbPP2AC6) to 65.57 kDa (HbPP2AA2), and their predicted isoelectric points ranged from 4.79 (HbPP2AC2-2) to 8.81 (HbPP2AB'η-5) (Table 1).

### Phylogenetic analysis of the HbPP2A proteins

To investigate the phylogenetic relationship among PP2A proteins, the 36 HbPP2A proteins and 25 AtPP2A proteins were used to construct a phylogenetic tree (S2 Table; Fig 1). All PP2A proteins were divided into five clusters. Four HbPP2AA (1–1, 1–2, 2, 3) and three AtPP2AA proteins (1, 2, 3) were clustered into the A subfamily, 14 HbPP2AB' (α, β, γ, ζ, η-1, η-2, η-3, η-4, η-5, θ-1, θ-2, κ-1, κ-2, μ) and nine AtPP2AB' proteins (α, β, γ, δ, ε, ζ, η, θ, κ) were grouped into the B' subfamily, six HbPP2AB" (α, β, δ, ε, TON2/FASS1-1, TON2/FASS1-2) and six

**Table 1. Detail information on putative *HbPP2A* genes.**

| Gene Name[a] | Gene Symbol | ProteinLength(aa) | Protein Mw(KD) | Protein pI | At_ortholog[b] | NCBI Accession |
|---|---|---|---|---|---|---|
| **A Subfamily** | | | | | | |
| HbPP2AA1-1 | scaffold1535_19427 | 588 | 65.46 | 4.84 | AtPP2AA1 (85%) | XM_021801654.1 |
| HbPP2AA1-2 | scaffold0700_10372 | 585 | 65.19 | 4.87 | AtPP2AA1 (82%) | XM_021784361.1 |
| HbPP2AA2 | scaffold0541_533719 | 588 | 65.57 | 4.93 | AtPP2AA2 (94%) | XM_021836026.1 |
| HbPP2AA3 | scaffold0064_1161940 | 474 | 52.65 | 4.94 | AtPP2AA3 (88%) | XM_021809592.1 |
| **B Subfamily** | | | | | | |
| HbPP2AB'α | scaffold1398_83140 | 495 | 56.87 | 6.35 | AtPP2AB'α (85%) | XM_021800237.1 |
| HbPP2AB'β | scaffold0341_796930 | 500 | 57.51 | 6.38 | AtPP2AB'β (83%) | XM_021826651.1 |
| HbPP2AB'γ | scaffold0052_1337195 | 545 | 61.58 | 8.58 | AtPP2AB'γ (80%) | XM_021805699.1 |
| HbPP2AB'ζ | scaffold0696_568009 | 544 | 61.88 | 8.49 | AtPP2AB'ζ (73%) | XM_021784212.1 |
| HbPP2AB'η-1 | scaffold0935_102261 | 534 | 61.27 | 7.02 | AtPP2AB'η (75%) | XM_021774761.1 |
| HbPP2AB'η-2 | scaffold0181_493108 | 526 | 60.00 | 6.98 | AtPP2AB'η (68%) | XM_021818399.1 |
| HbPP2AB'η-3 | scaffold0245_391232 | 525 | 60.40 | 8.58 | AtPP2AB'η (75%) | XM_021822042.1 |
| HbPP2AB'η-4 | scaffold0110_1891246 | 525 | 60.48 | 8.70 | AtPP2AB'η (75%) | XM_021813482.1 |
| HbPP2AB'η-5 | scaffold0110_1897177 | 367 | 41.97 | 8.81 | AtPP2AB'η (75%) | XM_021813366.1 |
| HbPP2AB'θ-1 | scaffold0935_109517 | 512 | 59.05 | 6.53 | AtPP2AB'θ (74%) | XM_021774805.1 |
| HbPP2AB'θ-2 | scaffold0181_504501 | 506 | 58.64 | 6.50 | AtPP2AB'θ (73%) | XM_021818400.1 |
| HbPP2AB'κ-1 | scaffold0428_120152 | 499 | 56.80 | 8.75 | AtPP2AB'κ (78%) | XM_021831463.1 |
| HbPP2AB'κ-2 | scaffold0163_1057839 | 517 | 58.74 | 6.81 | AtPP2AB'κ (63%) | XM_021817175.1 |
| HbPP2AB'μ | scaffold0342_789887 | 464 | 53.64 | 6.72 | NA | |
| HbPP2AB"α | scaffold0260_78377 | 458 | 52.30 | 5.04 | AtPP2AB"α(72%) | XM_021822605.1 |
| HbPP2AB"β | scaffold0260_119946 | 543 | 62.41 | 4.89 | AtPP2AB"β(78%) | XM_021822608.1 |
| HbPP2AB"δ | scaffold0388_752831 | 542 | 62.56 | 5.02 | AtPP2AB"δ(81%) | XM_021829276.1 |
| HbPP2AB"ε | scaffold1106_106821 | 538 | 61.90 | 4.98 | AtPP2AB"ε(80%) | XM_021795548.1 |
| HbPP2A-TON2/FASS1-1 | scaffold0285_1329839 | 477 | 54.78 | 4.97 | AtPP2A-TON2/FASS1 | XM_021823851.1 |
| HbPP2A-TON2/FASS1-2 | scaffold0566_104904 | 477 | 54.74 | 4.96 | AtPP2A-TON2/FASS1 | XM_021779285.1 |
| HbPP2AB55α/Bα-1 | scaffold1458_13182 | 522 | 58.20 | 5.58 | AtPP2AB55α/Bα (80%) | XM_021800879.1 |
| HbPP2AB55α/Bα-2 | scaffold4717_3485 | 435 | 48.92 | 5.72 | AtPP2AB55α/Bα (78%) | XM_021808472.1 |
| HbPP2AB55α/Bα-3 | scaffold2940_2756 | 392 | 43.80 | 5.60 | AtPP2AB55α/Bα (77%) | XM_021807218.1 |
| HbPP2AB55β/Bβ | scaffold1433_96782 | 516 | 57.53 | 5.68 | AtPP2AB55β/Bβ(83%) | XM_021800632.1 |
| **C Subfamily** | | | | | | |
| HbPP2AC1-1 | scaffold0190_958287 | 307 | 35.05 | 4.95 | AtPP2AC1(92%) | XM_021818854.1 |
| HbPP2AC1-2 | scaffold0073_601834 | 308 | 35.16 | 4.97 | AtPP2AC1(91%) | XM_021810306.1 |
| HbPP2AC2-1 | scaffold1810_29299 | 307 | 35.00 | 4.84 | AtPP2AC2(94%) | XM_021803653.1 |
| HbPP2AC2-2 | scaffold1148_186747 | 307 | 34.97 | 4.79 | AtPP2AC2(94%) | XM_021796522.1 |
| HbPP2AC4-1 | scaffold0440_568305 | 314 | 35.85 | 5.14 | AtPP2AC4(93%) | XM_021832072.1 |
| HbPP2AC4-2 | scaffold0494_114077 | 314 | 35.82 | 5.14 | AtPP2AC4(93%) | XM_021834615.1 |
| HbPP2AC4-3 | scaffold0200_1623588 | 314 | 35.56 | 5.20 | AtPP2AC4(93%) | XM_021819758.1 |
| HbPP2AC6 | scaffold0088_170958 | 280 | 31.56 | 5.05 | NA | XM_021811306.1 |

NA, not applicable;

[a]asterisk represents the genes identified in this work;

[b]shows the best Arabidopsis ortholog, and percentage in the brackets indicate similarity index.

AtPP2AB" proteins (α, β, γ, δ, ε, TON2/FASS1) were clustered into the B" subfamily, four HbPP2AB55 (α/Bα-1, α/Bα-2, α/Bα-3, β/Bβ) and two AtPP2AB55 proteins (α/Bα, β/Bβ) were grouped into the B55 subfamily, and eight HbPP2AC (1–1, 1–2, 2–1, 2–2, 4–1, 4–2, 4–3, 6) and five AtPP2AC proteins (1, 2, 3, 4, 5) were clustered into the C subfamily.

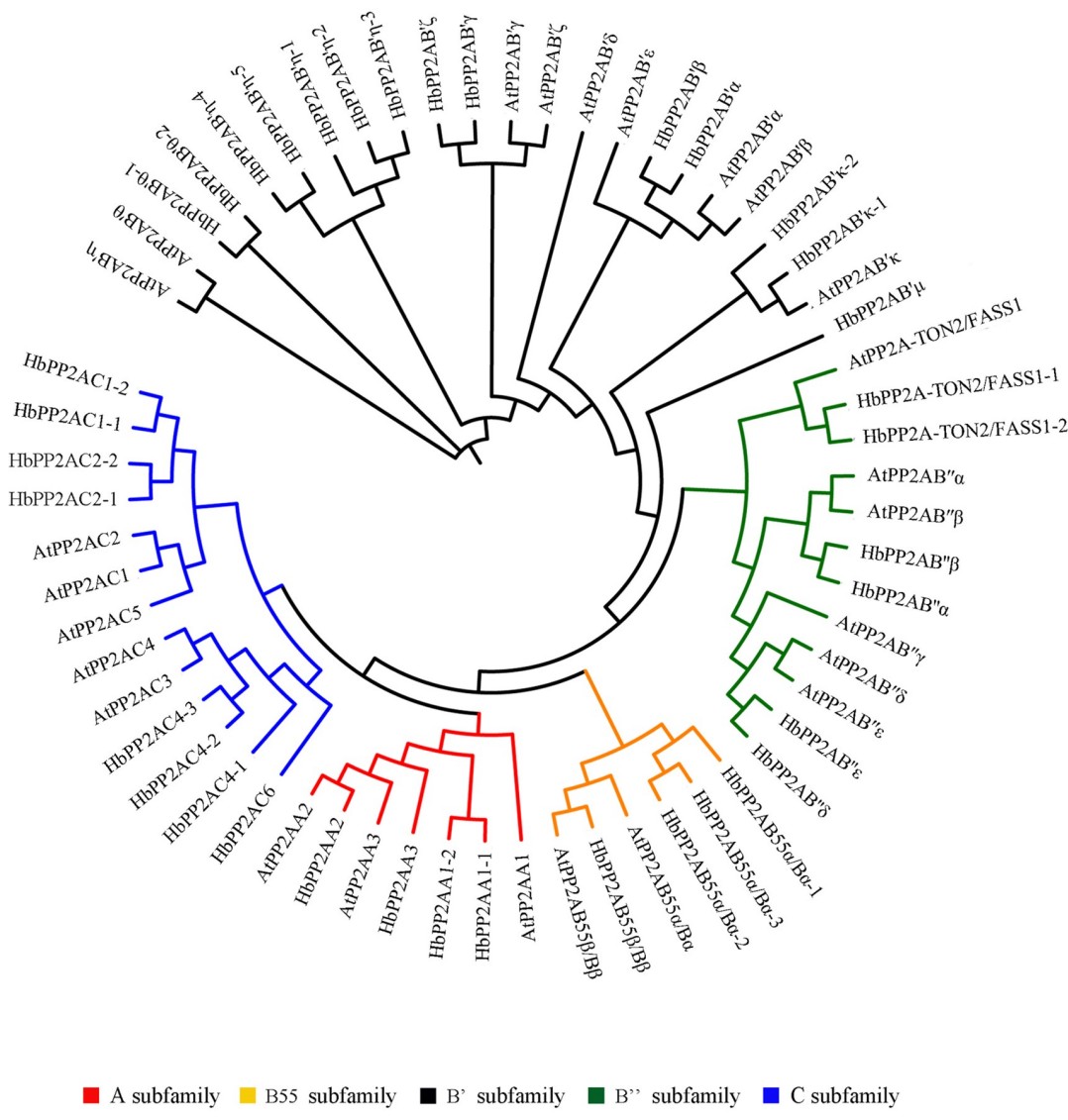

**Fig 1. Phylogenetic tree of the HbPP2As and AtPP2As.** All proteins were aligned by Multiple Sequence Alignment software and the unrooted phylogenetic tree was constructed using iTOL software.

## The structure of the HbPP2A genes

Exon-intron organization analysis of the 36 *HbPP2A* genes suggested that the intron numbers varied widely among the five subfamilies (Fig 2). For the *HbPP2AA* genes, the intron numbers ranged from 7 (*HbPP2AA3*) to 12 (*HbPP2AA2*). Most of the *HbPP2AB* genes had 11 introns, except for *HbPP2AB''α*, which contained nine introns. Three *HbPP2AC* genes (*4–1*, *4–2*, *4–3*) contained 10 introns, *HbPP2AC6* contained eight introns, while four *HbPP2AC* genes (*1–1*, *1–2*, *2–1*, *2–2*) only contained five introns. The *HbPP2AB55* genes possessed more introns, ranging from 8 (*HbPP2AB55α/Bα-3*) to 13 (*HbPP2AB55α/Bα-1* and *HbPP2AB55β/Bβ*), while the *HbPP2AB'* genes had fewer introns, ranging from 0 (*HbPP2AB'η-5*) to 2 (*HbPP2AB'η-1* and *HbPP2AB'β*).

A total of 20 conserved motifs were identified among the 36 HbPP2A proteins using MEME online software (Fig 3, S1 Fig). Motifs 1, 2, 3, 4, 5, 8, 9 were annotated as B56 domains;

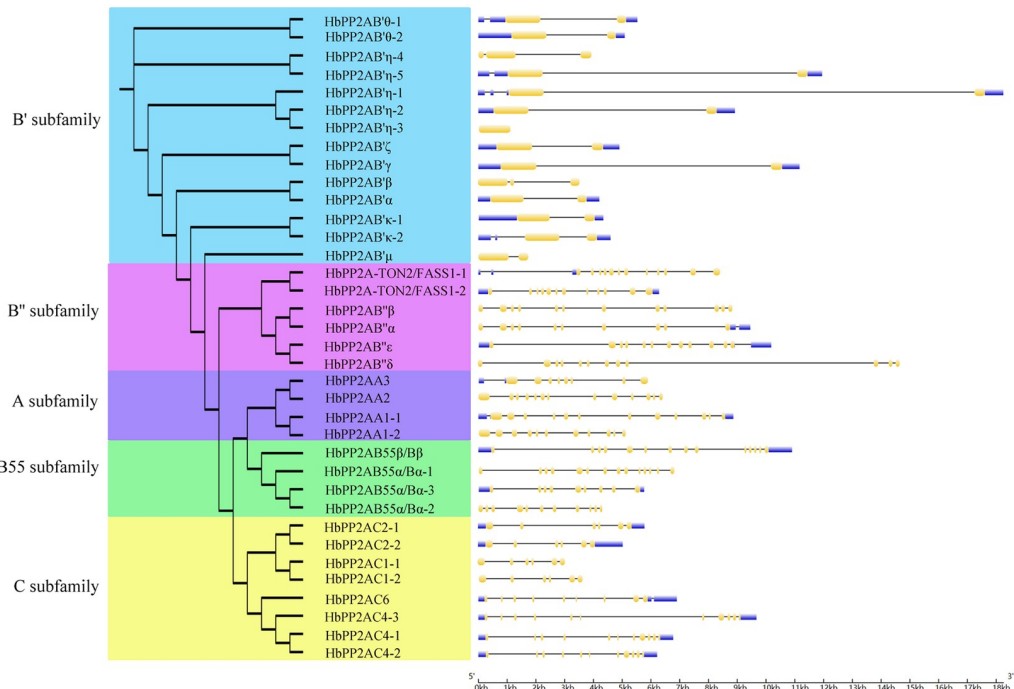

**Fig 2. The exon/intron organization of the *HbPP2As* family.** Thin black lines indicated introns. Thick yellow and thick blue boxes indicated coding sequence and untranslated region, respectively.

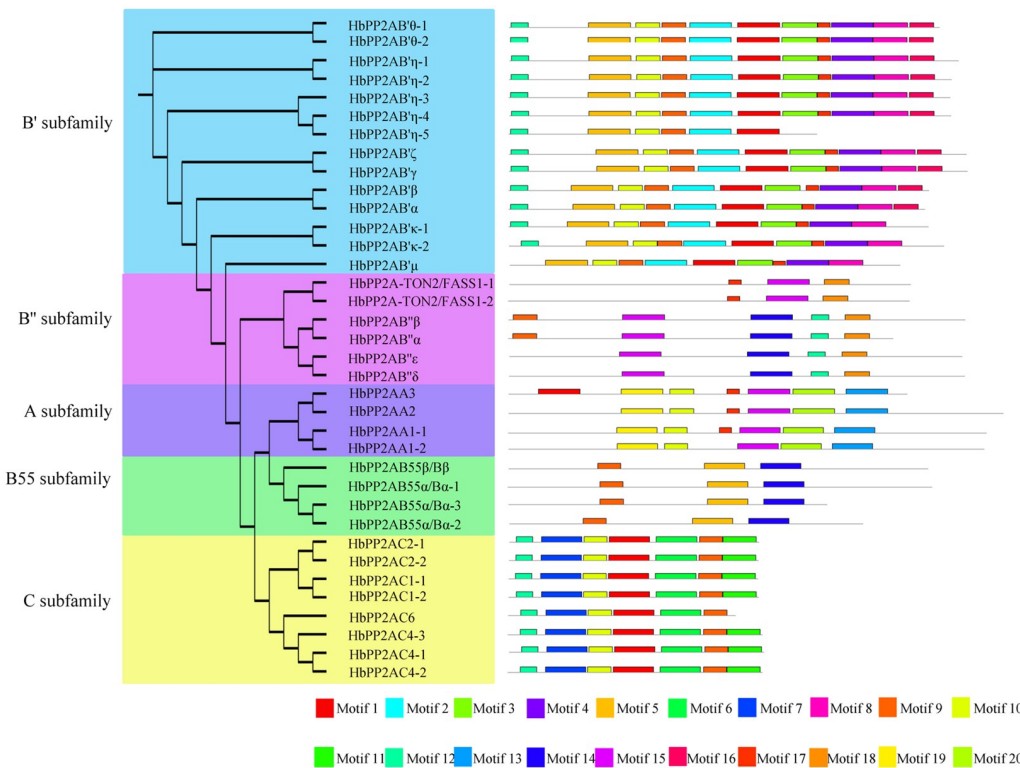

**Fig 3. Motif analysis of the HbPP2A family.** 20 conserved motifs were identified using the MEME program.

motif 18 was annotated as an EF-hand domain; motifs 13, 15, and 20 were annotated as HEAT domains; motif 7 was annotated as a serine/threonine phosphatase, but the remaining eight motifs (6, 10, 11, 12, 14, 16, 17, and 19) were not annotated in the InterProScan database. All HbPP2AB55 genes contained motifs 5, 9, 14; four HbPP2AB" genes (α, β, δ, ε) had motifs 12, 14, 15, and 18; and two HbPP2AB" genes (TON2/FASS1-1 and TON2/FASS1-2) had motif 15 and motif 18.

Analysis of 1,500 bp upstream sequence of *HbPP2As* identified several *cis*-acting elements related to hormones, stresses, developmental and metabolic regulation (Fig 4). For the hormone responsive category, the ethylene responsive element accounted for the largest proportion (ERE, 29/36), followed by abscisic acid responsive element (ABRE, 23/36), salicylic acid responsive elements (TCA-element, 17/36), jasmonic acid responsive elements (TGACG-motif, 16/36), auxin responsive elements (TGA-motif, 11/36; AuxRR-core, 3/36), gibberellin responsive elements (P-box, 7/36; TATC-box, 4/36; GARE-motif, 2/36). For the stress responsive category, 30 *HbPP2As* had wound responsive elements (WRE3, W-box, WUN-motif), 28 *HbPP2As* contained anaerobic induction element (ARE), 18 *HbPP2As* had drought-inducibility element (MBS), 17 *HbPP2As* possessed low-temperature responsiveness element (LTR), 12

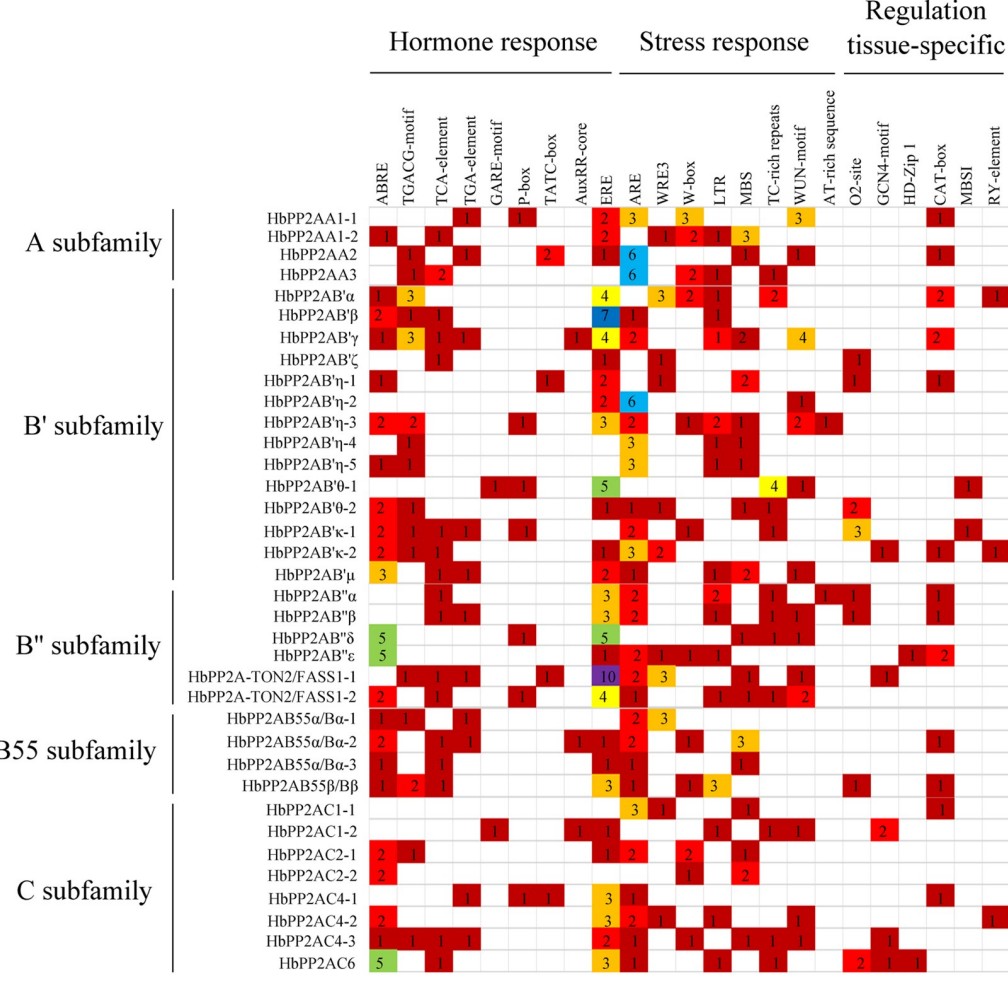

**Fig 4. *Cis*-acting elements in *HbPP2A* promoters.** 1.5 kb promoter sequences of 36 *HbPP2A* genes were analyzed using the PlantCARE database. The different colors represent the number of each cis-acting elements.

*HbPP2As* owned defense and stress responsiveness element (TC-rich repeats), two *HbPP2As* had elicitor-mediated activation element (AT-rich sequence). For the developmental and metabolic regulation category, CAT-box related to meristem expression was the most abundant element (13/36), followed by $O_2$-site involved in zein metabolism regulation (8/36), GCN4_motif involved in endosperm expression (5/36), RY-element involved in seed-specific regulation (3/36), HD-Zip1 involved in differentiation of the palisade mesophyll cells (2/36), MBSI involved in flavonoid biosynthetic genes regulation (2/36).

## The tissue-specific expression of *HbPP2A* genes

The published Illumina RNA-seq data representing seven parts (bark, female flower, laticifer, leaf, male flower, root, and seed) were used to assess the tissue-specific expression of *HbPP2A* genes. The 36 *HbPP2A* genes clearly showed a differential expression pattern among the seven tissues (Fig 5). All genes were expressed in at least one transcriptome data set, while a total of 15 genes, including *HbPP2AA1-1*, six *HbPP2AB′* (*α, γ, ζ, η-2, θ-1, κ-1*), three *HbPP2AB′* (*δ, ε, TON2/FASS1-2*), *HbPP2AB55β/Bβ*, and five *HbPP2AC* (*2–1, 2–2, 4–1, 4–2, 4–3*) genes, were expressed among all tested tissues. Among the tested organs or tissues, laticifer had the largest proportion of expressed *HbPP2A* genes (29/36). The seven undetected genes were *HbPP2AA2*, *HbPP2AB′θ-2*, *HbPP2AB′κ-2*, *HbPP2AB′η-5*, *HbPP2AB′μ*, *HbPP2AB55α/Bα-2*, and *HbPP2AB55α/Bα-3*. The transcripts of the undetected genes in laticifer were abundant in female flower and leaf for *HbPP2AA2*; bark and root for *HbPP2AB′θ-2*; female flower, leaf, male flower and root for *HbPP2AB55α/Bα-3*; and female flower, leaf, and root for *HbPP2AB55α/Bα-2*.

## The expression patterns of *HbPP2A* genes in latex upon tapping and ethrel application

The expression patterns of the 29 *HbPP2A* genes that were expressed in latex were further analyzed by qRT-PCR upon tapping and ethrel application (Fig 6, S3 Table). In comparison with virgin trees, two genes (*HbPP2AA1-1* and *HbPP2AB55α/Bα-1*) were up-regulated, seventeen genes were down-regulated and ten genes had not been affected in the tapped trees without ethrel application. In comparison with the tapped trees without ethrel application, *HbPP2AC2-2* was significantly up-regulated, twenty-one genes were down-regulated and eight genes had not been affected in the tapped trees with ethrel application. Of the down-regulated seventeen genes by tapping, fourteen genes were further significantly down-regulated by ethrel application.

## Discussion

As a group of eukaryotic serine/threonine protein phosphatases, PP2A shows high diversification in the subunits among species to adapt to the complex external environment [7]. Whole genome duplication (WGD) is an important evolutionary force that drives the expansion and diversification of gene families [23–24]. Booker and DeLong [4] showed that the expansion of PP2A subunit gene families in both flowering plants and animals is driven by WGD followed by non-random gene loss. Here, we identified a total of 36 HbPP2A proteins based on the latest version of the rubber tree genome [20]. In comparison with Arabidopsis PP2A proteins, the numbers of HbPP2A subunits are obviously expanded in each subfamily (Fig 1). Although the lack of a high-density genetic map impedes the deep exploration of the gene family expansion in rubber tree, several gene pairs (HbPP2AA1-1/HbPP2AA1-2; HbPP2AB'η-1/HbPP2AB'η-2/HbPP2AB'η-3/HbPP2AB'η-4/HbPP2AB'η-5; HbPP2AB'θ-1/HbPP2AB'θ-2; HbPP2AB'κ-1/HbPP2AB'κ-2; HbPP2A-TON2/FASS1-1/HbPP2A-TON2/FASS1-2;

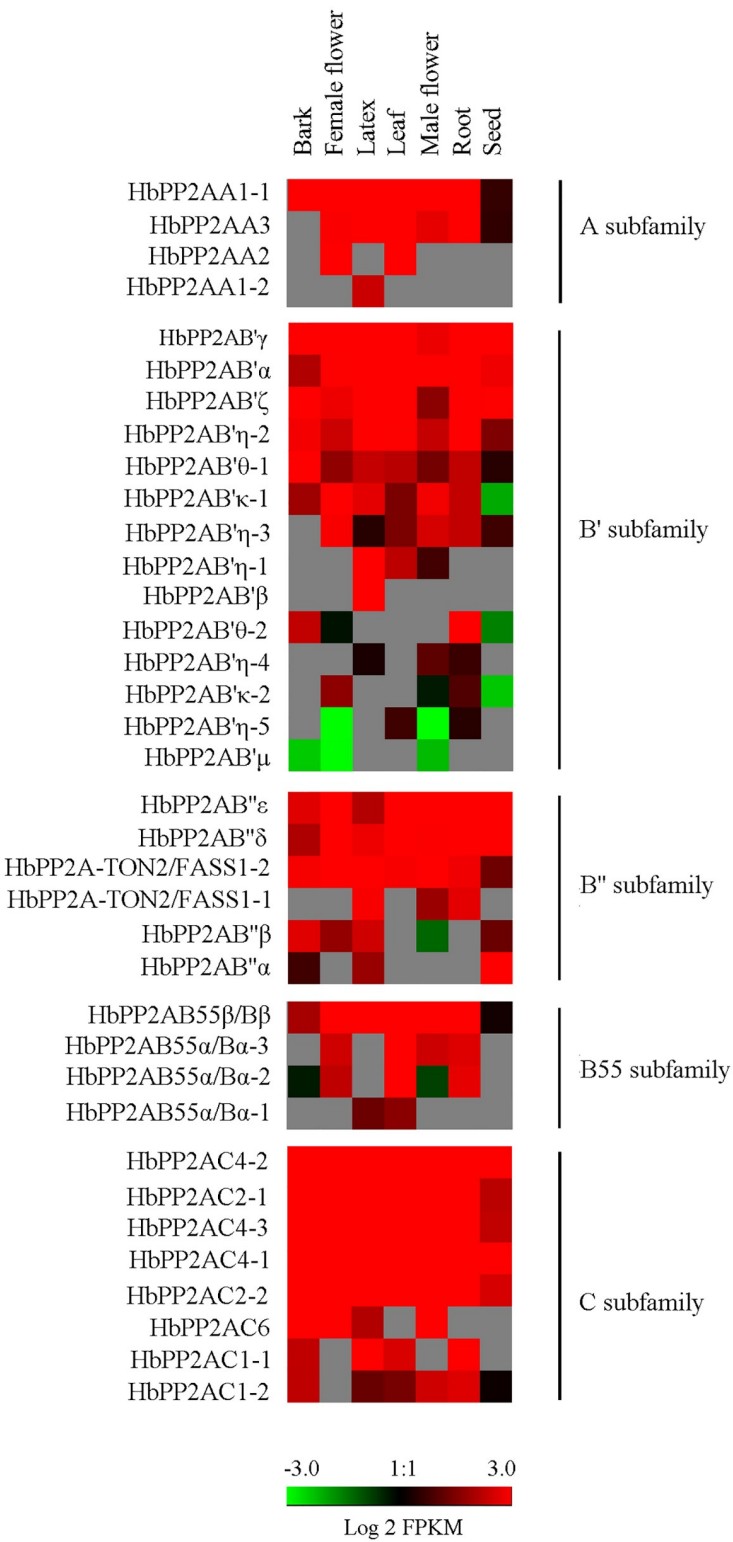

**Fig 5. Expression analyses of *HbPP2As* base on the published transcriptome data.** Red color indicated high expression while green represented very low expression. Grey showed no expression data detected. Bar represented the Log 2 FPKM value.

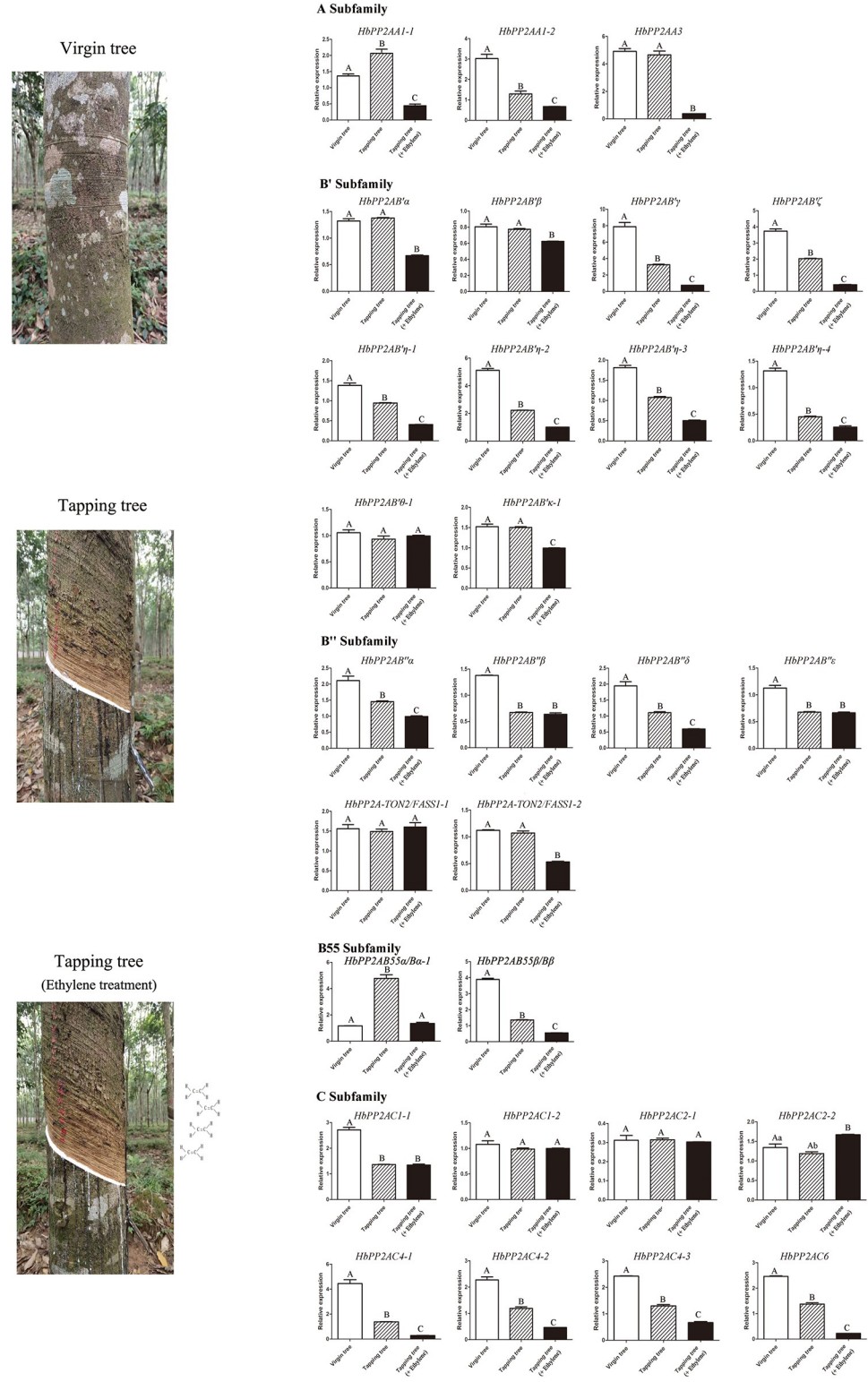

**Fig 6. Expression profiles of *HbPP2As* in latex upon ethrel treatment and tapping.** Error bars for qRT-PCR showed the standard deviation of three replicates. Different letters indicate statistically significant difference in the relative expression of each gene among different treatments. The capital letter represents p < 0.01 while lowercase represents p < 0.05.

HbPP2AB55α/Bα-1/HbPP2AB55α/Bα-2/HbPP2AB55α/Bα-3; HbPP2AC1-1/HbPP2AC1-2; HbPP2AC2-1/HbPP2AC2-2; HbPP2AC4-1/HbPP2AC4-2/HbPP2AC4-3) with obvious WGD features were identified (Table 1). Among these, the pair HbPP2AB'η-4/HbPP2AB'η-5 is further classified as being tandem duplicates based on the same-direction neighbours on the same scaffolds. The Ka/Ks ratio is an indicator of selective pressure [25–26]. In general, Ka/Ks > 1 represents positive selection, while Ka/Ks < 1 shows purifying selection. We showed that the Ka/Ks ratios of all duplicates are below one, suggesting that purifying selection plays a key role in their divergence (S4 Table).

The protein phosphorylated modification is manipulated by protein kinases and phosphatases [27–28]. Ethrel application increases the level of phosphorylation of a number of proteins in latex [16]. The enhanced protein phosphorylation in latex may be associated with MPK and CPK considering that four *HbMPK* and eight *HbCPK* genes in latex were up-regulated by ethrel application [17–19]. It may also be associated with HbPP2As as 21 of the 29 laticifer-expressed *HbPP2A* genes are significantly down-regulated upon ethrel application (Fig 6). The presence of ERE, a *cis*-acting element with a core sequence (TTTGAATT) that responses to ethylene [29–30] in the promoter regions of most *HbPP2As* confers their responses to ethrel application. In addition, 24 of 29 latex-expressed *HbPP2As* possess the wound responsive elements (WRE3, W-box, WUN-motif), suggesting their potential responsive to mechanical wounding. The *cis*-acting element WRE3 is absent from the 8 of 10 genes that have no response to tapping. Alternatively, the tapping-induced down-regulation of *HbPP2As* may mainly attribute to the wounding-induced endogenous ethylene since 14 of the 17 genes thatare down-regulated by tapping are further down-regulated by ethrel application. The wounding-induced ethylene formation is controlled by the regulation of 1-aminocyclopropane-1-carboxylic acid synthesis (ACS) [31]. The C-terminal domain of ACS can be phosphorylated by MPK and CPK and dephosphorylated by PP2A [32–34]. The ACS activity is significantly activated in the Arabidopsis *pp2aa1* mutant [35], suggesting that the ACS activity is positively related to the increased phosphorylation level of ACS by inhibiting AtPP2AA1. Here, the significant down-regulation of its ortholog, *HbPP2AA1-2*, may result in the production of endogenous ethylene by activating ACS activity.

The enhanced *in vitro* rubber biosynthesis of latex from tapped trees is closely associated with the up-regulation of the genes encoding the key enzymes for rubber biosynthesis by activating jasmonate signalling [14]. It may also be related to the changes in the key enzyme activity by phosphorylation modification. Available data shows that PP2As act as negative regulators for plant secondary metabolism. AtPP2AB'γ negatively regulates the methoxylation of hydroxylated indole glucosinolates and the formation of 4-hydroxyindol-3-yl-methyl glucosinolate [36]. AtP-P2AB''β acts as a posttranslational negative regulator of hydroxy-3-methyl glutaryl coenzyme A reductase activity [37]. With this respect, the tapping-caused down-regulation of *HbPP2As* may attribute to the enhanced rubber biosynthesis in tapped trees. Given the effects of ethrel application on enhanced phosphorylation of proteins in latex are mainly ascribed to the down-regulated expression of *HbPP2As*, the tapping-caused down-regulated expression of a large number of *HbPP2As* may result in the enhanced phosphorylation of proteins in latex from tapped trees. The effect of enhanced protein phosphorylation on rubber biosynthesis remains to be elucidated.

## Conclusions

Ethrel application induces the phosphorylation of a large number of proteins in the latex from rubber trees. The down-regulated expression of the most of the genome-wide identified HbPP2A genes may attribute to the enhanced phosphorylation of proteins in the latex from ethrel treated trees. The tapping-caused down-regulated expression of seventeen *HbPP2As*

may be related to the endogenous ethylene generation. The genome-wide identification of HbPP2A proteins will facilitate the regulation of the phosphorylized modification of rubber biosynthesis-related proteins in rubber tree.

## Supporting information

**S1 Fig. Conserved amino acid sequences of 20 motifs.**
(TIF)

**S1 Table. Primers used in this paper.**
(DOCX)

**S2 Table. Protein sequences used in this paper.**
(DOCX)

**S3 Table. qRT-PCR datas of *HbPP2A* genes expression.**
(XLSX)

**S4 Table. The Ka/Ks ratios and estimated absolute dates for the duplication events between the duplicated HbPP2As.**
(DOCX)

## Acknowledgments

We would like to thank other staff members of the Tian's lab for sample collection and data analysis.

## Author Contributions

**Conceptualization:** Jinquan Chao, Weimin Tian.

**Data curation:** Shuguang Yang, Xiaomin Deng.

**Funding acquisition:** Jinquan Chao, Weimin Tian.

**Methodology:** Zhejun Huang.

**Writing – original draft:** Jinquan Chao.

**Writing – review & editing:** Weimin Tian.

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
