## [Decision Letter · Decision Letter 0]

20 Dec 2019

PONE-D-19-32560

Genome-wide identification and expression analysis of the phosphatase 2A family in rubber tree (Hevea brasiliensis)

PLOS ONE

Dear Dr. Chao,

Thank you for submitting your manuscript to PLOS ONE. After careful consideration, we feel that it has merit but does not fully meet PLOS ONE’s publication criteria as it currently stands. Therefore, we invite you to submit a revised version of the manuscript that addresses all points raised during the review process, in particular toning down the conclusion that “moderately down-regulated expression of HbPP2As is crucial for the enhanced natural rubber biosynthesis in rubber trees”, as downregulation of PP2As at this point just correlates with enhanced rubber biosynthesis. To underscore a causal relationship, additional experiments are needed. In addition, qPCR data should be made available.

We would appreciate receiving your revised manuscript by 18 January 2020. To enhance the reproducibility of your results, we recommend that if applicable you deposit your laboratory protocols in protocols.io, where a protocol can be assigned its own identifier (DOI) such that it can be cited independently in the future. For instructions see: http://journals.plos.org/plosone/s/submission-guidelines#loc-laboratory-protocols

We look forward to receiving your revised manuscript.

Kind regards,

Veerle Janssens, Ph.D.

Academic Editor

PLOS ONE

Journal Requirements:

Reviewers' comments:

Reviewer's Responses to Questions

**Comments to the Author**

1. Is the manuscript technically sound, and do the data support the conclusions?

Reviewer #1: Yes

Reviewer #2: Partly

2. Has the statistical analysis been performed appropriately and rigorously? 

Reviewer #1: Yes

Reviewer #2: Yes

3. Have the authors made all data underlying the findings in their manuscript fully available?

Reviewer #1: Yes

Reviewer #2: No

4. Is the manuscript presented in an intelligible fashion and written in standard English?

Reviewer #1: Yes

Reviewer #2: Yes

5. Review Comments to the Author

Reviewer #1: This manuscript shows an analysis of the PP2A subunits in the rubber tree. The experimental part is mainly use of bioinformatics, but also some gene expression analysis is included. The identified subunits are compared with Arabidopsis data and other species for phylogenetic analysis and gene structures. The promoter regions of PP2A genes were analyzed and gave some interesting putative bindings sites for hormone and stress responses.

This is a well written manuscript with some interesting results.

Figure 6. For B55 Subfamily the gene names appear to be missing.

Reviewer #2: The publication describes the identification of Protein Phosphatase 2A family in rubber tree Hevea brasiliensis. Various bioinformatics tools have been applied to investigate the evolutionary background, gene structure, promoter cis-elements and transcriptional activity of the different PP2A subunits. In addition to the bioinformatics analysis, the authors present qPCR data of the transcriptional activity of the PP2A subunits in virgin trees, in tapped trees and in tapped trees after ethrel treatment. The analysis and experiments have been described with appropriate detail and are of good technical quality.

The discussion offers interesting conclusions about the role of protein phosphorylation in the regulation of rubber biosynthesis. The authors bring up that increased protein phosphorylation and transcriptional upregulation of several protein kinases are have been observed after tapping and ethrel application. The manuscript provides evidence that this is coincided with decreased abundance of PP2A transcripts. However, based on the evidence presented in the manuscript, I feel that the main conclusion “moderately down-regulated expression of HbPP2As is crucial for the enhanced natural rubber biosynthesis in rubber trees” is not fully justified. The study does not provide experimental evidence, that the moderate down regulation of PP2As is necessary for the enhanced rubber biosynthesis. Altogether, the manuscript presents interesting findings and new data about PP2As in Hevea brasiliensis but the authors should rephrase the conclusions of the manuscript.

Other comments:

The labeling in the Figure 6 is misleading as the trees were treated with ethrel and not with ethylene.

Although the manuscript is written in good English, I feel that careful editing of the language would enhance the readability of the manuscript.

According to PlosOne policy, the qPCR data presented in the manuscript should be made available for the reader.

6. PLOS authors have the option to publish the peer review history of their article (what does this mean?). If published, this will include your full peer review and any attached files.

Reviewer #1: No

Reviewer #2: No

---

## [Author Response · Author response to Decision Letter 0]

7 Jan 2020

Reviewer 1:

1 Figure 6. For B55 Subfamily the gene names appear to be missing.

Answer: Thank you for your kind suggestion. We have added " HbPP2AB55α/Bα-1" and " HbPP2AB55β/Bβ" in the revised Fig 6 in new version. 

Reviewer 2:

1 Have the authors made all data underlying the findings in their manuscript fully available?

Answer: Thank you for your kind suggestion. We have supplied the qRT-PCR data as attachment (S3 Table).

2 However, based on the evidence presented in the manuscript, I feel that the main conclusion “moderately down-regulated expression of HbPP2As is crucial for the enhanced natural rubber biosynthesis in rubber trees” is not fully justified. The study does not provide experimental evidence, that the moderate down regulation of PP2As is necessary for the enhanced rubber biosynthesis. Altogether, the manuscript presents interesting findings and new data about PP2As in Hevea brasiliensis but the authors should rephrase the conclusions of the manuscript.

Answer: Thank you for your kind suggestion. In the revised version, we suggest that down-regulated expression of HbPP2As may contribute to the enhanced phosphorylation of proteins by ethrel application. Whether the protein phosphorylized modification on the activity of rubber biosynthesis-related proteins remains elucidated. 

3 The labeling in the Figure 6 is misleading as the trees were treated with ethrel and not with ethylene.

Answer: Thanks. We have modified the labeling in the revised Fig 6 in new version.

4 Although the manuscript is written in good English, I feel that careful editing of the language would enhance the readability of the manuscript.

Answer: Thanks. We have re-checked the MS thoroughly in the revised version. 

5 According to PlosOne policy, the qPCR data presented in the manuscript should be made available for the reader.

Answer: Thanks. We have supplied the qRT-PCR data as a attachment (S3 Table).

---

## [Editor Report · Decision Letter 1]

10 Jan 2020

Genome-wide identification and expression analysis of the phosphatase 2A family in rubber tree (Hevea brasiliensis)

PONE-D-19-32560R1

Dear Dr. Chao,

We are pleased to inform you that your manuscript has been judged scientifically suitable for publication and will be formally accepted for publication once it complies with all outstanding technical requirements.

With kind regards,

Veerle Janssens, Ph.D.

Academic Editor

PLOS ONE

Additional Editor Comments (optional):

The authors have adequately addressed the reviewers' and editor's comments
---

## [Editor Report · Acceptance letter]

24 Jan 2020

PONE-D-19-32560R1 

Genome-wide identification and expression analysis of the phosphatase 2A family in rubber tree (*Hevea brasiliensis*) 

Dear Dr. Chao:

I am pleased to inform you that your manuscript has been deemed suitable for publication in PLOS ONE. Congratulations! Your manuscript is now with our production department. 

With kind regards,

on behalf of

Prof. Veerle Janssens 

Academic Editor

PLOS ONE